# DREAMGUIDER: IMPROVED TRAINING FREE DIFFUSION-BASED CONDITIONAL GENERATION

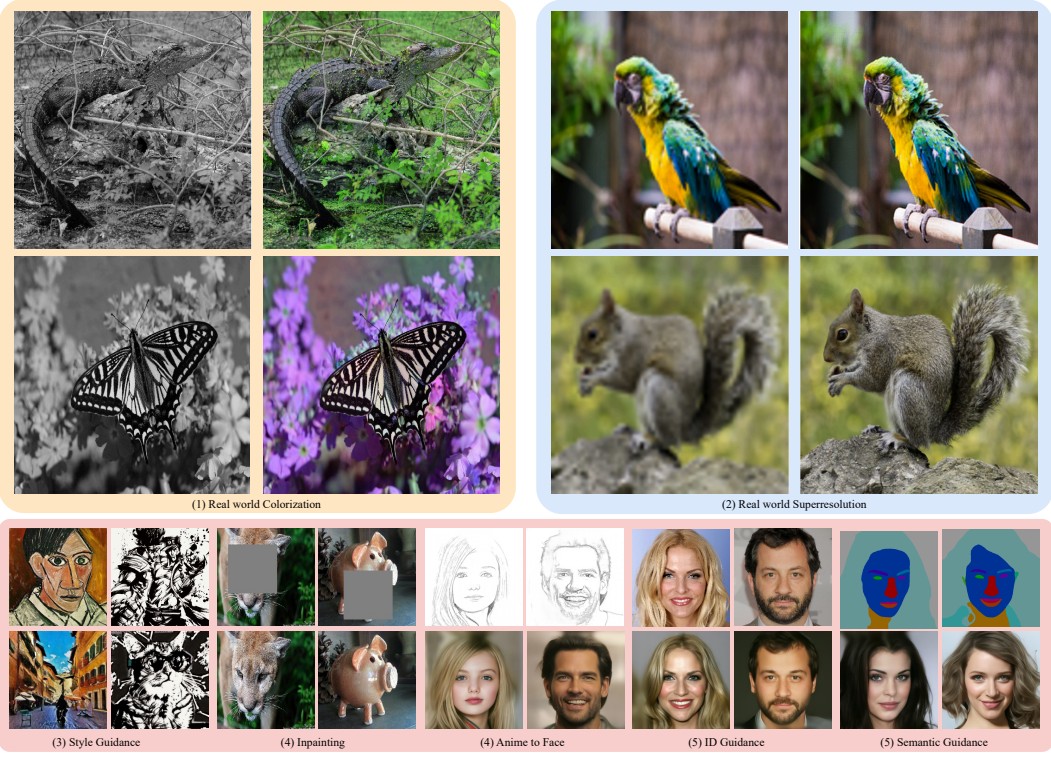

Figure 1: An illustration of the different applications of our method. We utilize a pretrained diffusion model to generate images satisfying a predefined condition without backpropagation through the diffusion UNet or any hand-crafted parameter tuning. We present results on (1) Real-world colorization, (2) Real-world super-resolution, (3) Style-guided Text-to-Image Generation, (4) Inpainting, (5) Sketch-to-Face, (6) Face ID Guidance, and (7) Face Semantics-to-Face synthesis.

## ABSTRACT

Diffusion models have emerged as a formidable tool for training-free conditional generation. However, a key hurdle in inference-time guidance techniques is the need for compute-heavy backpropagation through the diffusion network for estimating the guidance direction. Moreover, these techniques often require handcrafted parameter tuning on a case-by-case basis. Although some recent works have introduced minimal compute methods for linear inverse problems, a generic lightweight guidance solution to both linear and non-linear guidance problems is still missing. To this end, we propose Dreamguider, a method that enables inference-time guidance without compute-heavy backpropagation through the diffusion network. The key idea is to regulate the gradient flow through a time-varying factor. Moreover, we propose an empirical guidance scale that works for a wide variety of tasks, hence removing the need for handcrafted parameter tuning. We further introduce an effective lightweight augmentation strategy that significantly boosts the performance during inference-time guidance. We present experiments using Dreamguider on multiple tasks across multiple datasets and models to show the effectiveness of

the proposed modules. To facilitate further research, we will make the code public after the review process.

# 1 INTRODUCTION

Generative modeling utilizing Denoising Diffusion Probabilistic Models (DDPMs) Sohl-Dickstein et al. (2015); Ho et al. (2020); Dhariwal & Nichol (2021); Song et al. (2021b) has massively improved over the past few years. Multiple works have extended the use of diffusion models for text-to-image synthesis Balaji et al. (2022); Rombach et al. (2021); Saharia et al. (2022b), 3D synthesis Poole et al. (2022); Jun & Nichol (2023), video generation Ho et al. (2022); Blattmann et al. (2023); Wu et al. (2023a), as well as for conditioning to solve inverse problems. Moreover, like conditional generative adversarial networks (GANs)Goodfellow et al. (2020); Arjovsky et al. (2017), DDPMs can be adapted to tasks based on a labels Rombach et al. (2021); Dhariwal & Nichol (2021) or visual prior-based conditioning Saharia et al. (2022a). However, like conditional GANs Wang et al. (2018); Radford et al. (2015), DDPMs also need to be trained with annotated pairs of labels and instructions to obtain satisfactory results. This poses a limitation in many cases where there is a lack of paired data to train large diffusion models. Due to this reason, there has been recent interest in models that can perform conditional generation without the need for explicit training Yu et al. (2023); Chan et al. (2016); Nguyen et al. (2017); Graikos et al. (2022).

Progressing towards this direction is prior research in plug-and-play models. First introduced in Nguyen et al. (2017), the initial research on plug-and-play models Nguyen et al. (2017); Graikos et al. (2022) enabled conditional sampling from GANs trained with unlabeled data. For this, a pre-trained classifier Simonyan & Zisserman (2014); Hossain et al. (2019) or a captioning model was used to estimate the deviation between the GAN-generated image and a given label, and based on this deviation, the GAN input noise was modulated until the generated sample satisfied the given text or class label. A similar approach that has been attempted for diffusion models to facilitate conditional sampling from unconditional diffusion models is classifier guidance Dhariwal & Nichol (2021); Graikos et al. (2022), where a noise-robust classifier is trained along with the diffusion model to guide the sampling towards a particular direction. However, classifier guidance brings in the computational costs of training a classifier, which is often undesirable. Some recent works have performed conditional generation without explicit training for the condition by utilizing the implicit guidance capabilities of the diffusion model Chung et al. (2023b); Yu et al. (2023); Nair et al. (2023); Bansal et al. (2023); Chung et al. (2023a). Diffusion posterior sampling (DPS) Chung et al. (2023b) proposed a technique of using an $L_2$ norm-based loss function to solve linear inverse problems using unconditional diffusion models. However, DPS often requires a large number of sampling steps for photorealistic results. Freedom Yu et al. (2023), yet another work, proposed the use of general loss functions during sampling to achieve training-free conditional sampling. Some variants of DPS have also been proposed in the literature Song et al. (2023). All the aforementioned loss-guided posterior sampling techniques involve a guidance function at each timestep that requires backpropagation through the diffusion UNet. Recently, He et al. (2023) proposed Manifold Preserving Guided Diffusion Models (MGD) that remove the need for backpropagating through the diffusion U-Net by performing a gradient descent with respect to the Minimum Mean Square Error (MMSE). Although MGD He et al. (2023) works remarkably well for linear tasks that require more guidance towards the start of the guidance process, it may fail in some tasks where guidance happens earlier, for example, face semantics-to-image and sketch-to-image, where stronger guidance is required from a much earlier stage. Moreover, like Yu et al. (2023); Nair et al. (2023), MGD also requires a case-by-case handcrafted parameter. Hence, a generic lightweight method that works well for both linear and non-linear guidance functions is still missing. Moreover, the need to find a handcrafted guidance parameter on a case-by-case basis still remains an open challenge.

In this paper, we introduce a new framework that can adaptively perform zero-shot generation using diffusion models without the need for any manual intervention by the user. We found a rather simple fix to the problem during the initial timesteps of diffusion, i.e., by utilizing the gradient with respect to the diffusion output noise in initial steps of inference. Combined with the guidance with respect to the MMSE estimate, we found that the combination generalizes well to tasks that require guidance at very early stages of guidance. Figure 2 presents the visualization of our approach over existing works present in the literature. Utilizing the correction term along with the correction with respect to the MMSE estimate significantly boosts the performance in non-linear tasks. We

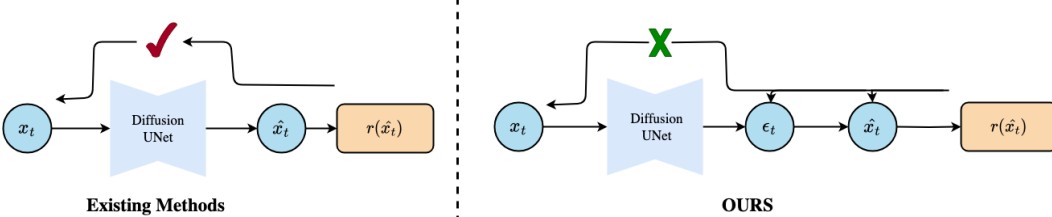

Figure 2: An illustration of the difference between the existing method and our method. Existing works backpropagate through the diffusion network to perform guidance at each timestep, whereas we find the gradients with respect to the MMSE estimate and the predicted noise based on the timesteps, thereby bypassing the expensive backpropagation operation.

Table 1: Table illustrating the difference over existing methods performing inference-time guidance.

| Method | Zeroth order | Linear Tasks | Non-Linear Tasks | Automatic scaling |
|---|---|---|---|---|
| DPS Chung et al. (2023a) | ✗ | ✓ | ✗ | ✗ |
| πGDM Song et al. (2022) | ✗ | ✓ | ✗ | ✗ |
| Freedom Yu et al. (2023) | ✗ | ✗ | ✓ | ✗ |
| MGD He et al. (2023) | ✓ | ✓ | ✗ | ✗ |
| OURS | ✓ | ✓ | ✓ | ✓ |

present the corresponding results in Section 5. Moreover, we treat the energy-based inference-time guidance Chung et al. (2023b); Yu et al. (2023) as a stochastic gradient optimization of the MMSE estimate and the noise present in the image. This formulation enabled us to leverage recent research in parameter-free learning Defazio & Mishchenko (2023); Ivgi et al. (2023) to develop a dynamic step size schedule. This step size adjusts itself adaptively based on the initial noise seed input of the diffusion model and guidance functions, hence removing the need for manual parameter tuning for inference-time guidance. Moreover, motivated by the effectiveness of differentiable augmentations while training GANs Zhao et al. (2020), we found that utilizing multiple levels of matching differentiable augmentations to the MMSE estimate and guidance reference significantly improves the sampling quality, enabling very high-quality sampling with a low number of guidance steps. We present an overview of the different applications of our method in Figure 1 and an illustration of the difference of dreamguider with existing methods in Table 1. Namely, we present results using Stable Diffusion Rombach et al. (2021), unconditional diffusion models released by Nichol & Dhariwal (2021) for $256 \times 256$ guidance, and class-conditional diffusion models for high-resolution $512 \times 512$ conditional synthesis. The different functionalities of Dreamguider are tabulated in Figure 2.

We present experiments on publicly released models on generic images, face images, and stable diffusion to show the relevance of our method. We focus on the tasks of (1) Inpainting, (2) Super-resolution, (3) Colorization, (4) Gaussian Deblurring, (5) Semantic label-to-image generation, (6) Face sketch-to-image, (7) ID guidance and identity generation, and beat existing benchmarks that utilize diffusion models for these tasks, obtaining a significant boost in performance over existing methods leveraging loss-guided models. To summarize, our contributions are:

- We propose a zeroth-order loss-guided diffusion guidance that is applicable to both linear inverse problems and non-linear inverse problems.
- We remove the need for a manually tuned guidance scale for classifier guidance by proposing a scaling function that works for a wide variety of tasks.
- We propose a time-varying guidance scale for improving sampling quality.
- We propose a differentiable augmentation strategy to improve sampling quality.

## 2 BACKGROUND

### 2.1 TRAINING-FREE CONDITIONAL SAMPLING USING DIFFUSION MODELS

Recently, there has been a rise in multiple works that propose utilizing unconditional diffusion models for conditional sampling Nair et al. (2023); Bansal et al. (2023); Chung et al. (2023c); Kawar et al. (2022). The earlier works proposed solving linear inverse problems using diffusion

models with the help of priors dependent on the inverse transform of degradation. Recently, diffusion posterior sampling Chung et al. (2023b) considered the degradation to be conditioned on a Gaussian distribution given any intermediate timestep and derived an $L_2$ norm-based regularization at each intermediate timestep to solve for linear inverse problems. Recent works such as Freedom Yu et al. (2023) explored an energy-based perspective and extended guidance to non-linear functions using general loss functions. Universal diffusion guidance Aggarwal et al. (2018) extended this guidance process to stable diffusion and improved the performance by using forward-backward guidance. More recent works, such as manifold-guided diffusion He et al. (2023), further proposed to constrain the manifold space by projecting for the latent space alone.

## 2.2 PERTURBED MARKOVIAN KERNEL FOR DIFFUSION TRANSITION

Let us assume that $r(x_t, y)$ gives a measure of the distance between an intermediate $x_t$ and the condition $y$ and is a positive bounded function. Hence, in the reverse process, the diffusion trajectory should proceed through distributions with a higher probability of being closer to the desired cases. We model these trajectory intermediate distributions with

$$\hat{p}(x_t) = p(x_t)r(x_t, y). \tag{1}$$

Dickenson et al. Sohl-Dickstein et al. (2015) first proposed the use of Markovian kernels to estimate the distribution of diffusion intermediates. Specifically, given the state $x_t$ at the equilibrium of the training process for a diffusion model, the intermediate of a diffusion model at a time instant, the distribution at a timestep $t - 1$ can be estimated as

$$p(x_{t-1}) = \int p(x_t)p_\theta(x_{t-1}|x_t)dx_t. \tag{2}$$

As we know, the kernel $p(x_{t-1}|x_t)$ is a Gaussian distribution whose mean can be estimated using the diffusion UNet and $x_t$. To estimate a perturbed kernel $\hat{p}(x_{t-1}|x_t)$, the perturbed distribution is

$$p(x_{t-1})r(x_{t-1}, y) = \int r(x_t, y)p(x_t)\hat{p}_\theta(x_{t-1}|x_t)dx_t. \tag{3}$$

By merging the constant terms in the transition into the normalization factor, the transition step is

$$\hat{p}_\theta(x_{t-1}|x_t) = p_\theta(x_{t-1}|x_t)r(x_{t-1}, y). \tag{4}$$

The proof is given in the supplementary material. Hence, we can see that rather than considering a Gaussian posterior, as in DPS Chung et al. (2023b), any distance or loss function can be used. Similarly, one other valid transition step of the perturbed process is

$$\hat{p}_\theta(x_{t-1}|x_t) = p_\theta(x_{t-1}|x_t)\frac{r(x_{t-1}, y)}{r(x_t, y)}, \tag{5}$$

which adopts the notion of reciprocal distance from the previous timestep.

## 2.3 INFERENCE-TIME GUIDANCE OF DIFFUSION MODELS

For conditional generation tasks using an unconditional diffusion model, ideally, the model would predict intermediates closer to the condition. The formulation can be seen in terms of transition probabilities. Consider a pretrained unconditional diffusion model on a specific domain. The problem at hand needs to guide the diffusion model during inference time conditioned with a condition $y$. Dhariwal et al. Dhariwal & Nichol (2021) proposed a general strategy to perform this by conditioning on the condition $y$ and finding the resultant marginal distribution

$$p(x_t|x_{t+1}, y) = p(x_t|x_{t+1})p(y|x_t). \tag{6}$$

By assuming the distribution $p(y|x_t)$ has much lower curvature compared to $p(x_t|x_{t+1})$, considering the marginal distribution close to $x_t$,

$$\log p(y|x_t) = (x_t - \mu)\nabla_{x_t}\log p(y|x_t), \tag{7}$$
$$g = \nabla_{x_t}\log p(y|x_t).$$

Plugging back to $\log(p(x_t|x_{t+1}, y))$,

$$\log(p(x_t|x_{t+1}, y)) = (x - \mu - \Sigma g)^T \Sigma^{-1}(x - \mu - \Sigma g) + C, \tag{8}$$
$$p(x_t|x_{t+1}, y) \sim N(\mu + \Sigma g, \Sigma).$$

Hence, the reverse sampling equation becomes,

$$x_{t-1} = \frac{1}{\sqrt{\alpha_t}} \left( x_t - \frac{1-\alpha_t}{\sqrt{1-\bar{\alpha}_t}} \epsilon_\theta(x_t) \right) + \sigma_t \epsilon + \Sigma \frac{dr(x_{t-1}, y)}{dx_{t-1}}, \epsilon \sim \mathcal{N}(0, I). \tag{9}$$

## 2.4 SHORTCOMINGS OF THE EXISTING METHODS

Although the energy-based guidance theory supports guidance as a function of the current latent estimate, almost all loss-based guidance techniques derive the distance function as a function of $x_t$ rather than $x_{t-1}$ and derive the gradient based on the previous sample. Although this approach works for many tasks, it requires backpropagating through the neural network and modeling the score function for the guidance correction term. This limits the use of classifier guidance since existing diffusion architectures that produce photorealistic results are often very bulky. One can see why the existing framework utilizes the derivative with respect to the previous sample works by taking a better look at Equation (5). As we can see, a reciprocal distance over the previous timestep diffusion latent $x_t$ is a perfectly valid distance guidance function. In the next section, we elaborate on Dreamguider.

## 3 PROPOSED METHOD

Suppose $x_{t-1}$ denotes the current step and $x_t$ denotes the previous step in the inference process of the diffusion module. As mentioned in the previous section, existing works utilize the derivative with respect to the previous step for guidance; one reason for this is to use an off-the-shelf auxiliary distance function on the MMSE estimate at each step $\hat{x}_t$, which enables the use of general functions defined on image space for guidance. Here, the MMSE estimate is defined as

$$\hat{x}_t = \frac{x_t - \sqrt{1-\bar{\alpha}_t}\epsilon_\theta(x_t)}{\sqrt{\bar{\alpha}_t}}, \tag{10}$$

where $\bar{\alpha}$ denotes the variance schedule of the diffusion process and $\epsilon_\theta(x_t)$ is the noise estimated by the network. One other observation to note is that finding the derivative with respect to the current step requires finding $\hat{x}_{t-1}$, which again requires an additional propagation through the diffusion network. Hence, the dilemma of backpropagating through the UNet for guidance still remains unresolved.

### 3.1 TIME VARIANT CLASSIFIER GUIDANCE

We found a simple yet effective solution for this dilemma; if we take a look at the ODE estimate at each step proposed by Song et al. Song et al. (2021a). Hence, rather than perturbing the Gaussian kernel at each timestep, we perturb the components $\hat{x}_t$ and $\epsilon_\theta(x_t)$ by a small amount. Specifically, we perform the following operations:

$$\hat{x}_t = \hat{x}_t - c\Sigma \frac{dr(\hat{x}_t, y)}{d\hat{x}_t}, t > t_0$$

$$\epsilon_\theta(x_t) = \epsilon_\theta(x_t) - d\Sigma \frac{dr(\hat{x}_t, y)}{d\epsilon_\theta(x_t)}, t < t_0$$

$$x_{t-1} = \frac{1}{\sqrt{\alpha_t}} \left( x_t - \frac{1-\alpha_t}{\sqrt{1-\bar{\alpha}_t}} \epsilon_\theta(x_t) \right) + \sigma_t \epsilon - c_t \Sigma \frac{dr(\hat{x}_t, y)}{d\hat{x}_t} - d_t \Sigma \frac{dr(\hat{x}_t, y)}{d\epsilon_\theta(x_t)} \tag{11}$$

where $r(\hat{x}_t, y)$ is a non negative distance function that measures the distance between the MMSE estimate and condition, $\Sigma$ is the variance of the latent estimate at each timestep as in Equation (8). Please note that we perform a double descent here. The intuition behind the double descent is that performing descent on one of the components, say $\hat{x}_t$, guides effectively at the end of the diffusion process where $\alpha_{t-1}$ is one and vice versa. Hence, during the guidance with the gradient w.r.t. $\hat{x}_t$, the maximum component of shift that happens to the sample is when we consider the flow of this correction through $\hat{x}_t$. Hence, we define the value as the maximum component of $x_{t-1}$ present in $\hat{x}_t$.

$$c_t = c\sqrt{\alpha_{t-1}}. \tag{12}$$

Similarly, we define $d_t$ as the maximal component of $\epsilon_\theta(x_t)$ in $x_{t-1}$. Hence,

$$d_t = -d.\frac{1-\alpha_t}{\sqrt{\alpha_t}\sqrt{1-\bar{\alpha_t}}}. \tag{13}$$

Hence, this term gives efficient guidance at all timesteps, bypassing the guidance at the later timesteps alone as in MGD He et al. (2023). In the following section, we proceed to propose an effective empirical estimate for $c$ and $d$ that works for a wide range of tasks.

## 3.2 A Gradient-Dependent Scaling Factor Estimate

Recently, Distance over Gradients (DOG) Ivgi et al. (2023) was proposed as an effective parameter-free dynamic step size schedule for SGD problems. Given any Stochastic Gradient Descent (SGD) optimization problem, the Distance over Gradient works as an effective learning rate. Recent works Wu et al. (2023b) have found the diffusion process as a stochastic optimization problem and have derived an SGD-based interpretation of the diffusion sampling process. Hence, inspired by both of these works, we attempted an empirical guidance estimate of the form:

$$\gamma_t = \begin{cases} \frac{1e^{-5}}{\sqrt{g_T^2}}, & \text{if } t = T \\ \frac{\max_{i>t}|f_i - f_T|}{\sqrt{\Sigma_{i=i}^T g_t^2}}, & \text{otherwise} \end{cases} \tag{14}$$

where $g_t$ is the gradient of the loss function as defined in the equation, $f_t$ can be any of $\hat{x}_t, x_t, \epsilon_\theta(t)$ at timestep $t$ and $f_0$ is the initial estimate of $f_t$. We noticed that this empirical estimate works well for the first-order sampling involving DPS Chung et al. (2023b) as well. We illustrate more results on the effect of this plug-in value for different cases in the appendix. Hence, utilizing Equation (14), we estimate $c$ and $d$ accordingly by substituting $f_i$ as $\hat{x}_t$ and $\epsilon_\theta(x_t)$

## 3.3 Differential Augmentation Classifier Guidance

A common practice while performing classifier guidance to augment diffusion models with specific regularization for guidance is to use the noisy estimate at timestep $t$ and utilize it to compute the loss function to regularize the current prediction. However, in many cases, such guidance can give results with artifacts and color shifts, as portrayed in Figure 3 and Figure 5, due to excessive guidance or insufficient guidance at intermediate timesteps that shift the results off manifold or cause color shifts. One effective solution for this is to imitate different versions of artifacts or color shifts on both the source image and the target image and utilize these augmented versions for a boost in performance. Hence, to perform guidance with a much more robust guidance loss, we introduce DiffuseAugment, an augmentation strategy for diffusion guidance during inference time. Specifically, given an intermediate sample $x_t$ and condition $y$, we augment $\hat{x}_t$ and $y$ with differentiable augmentations denoted by

$$\hat{x}_t^{aug}, y^{aug} = T(\hat{x}_t^{aug}, y^{aug}). \tag{15}$$

We choose three different types of augmentations for $T$ comprising random cutouts, random translations, and color saturations. Please note that the augmentation of $y$ is dependent on the input signal. For label-based conditioning such as identity or text, we do not perform augmentation for $y$. For image space augmentations, we augment $y$ with the same random augmentation as that of $x$. While computing the effective loss, we find the average across all augmentations. We find that DiffuseAugment significantly boosts the sampling fidelity and quality of the reconstructed image. We present these results in Section 5.

## 4 Experiments

Since our method comprises both linear and non-linear inverse tasks, for linear inverse tasks, we follow DPS and evaluate our method utilizing two different benchmarks: (1) ImageNet Deng et al. (2009) and (2) CelebA Liu et al. (2015). For non-linear tasks, we follow Freedom and evaluate using the CelebA dataset. For linear tasks, we evaluate our method quantitatively for Super-resolution ($\times 4$), Colorization, Inpainting (Box), and Gaussian deblurring tasks. For non-linear tasks, we evaluate

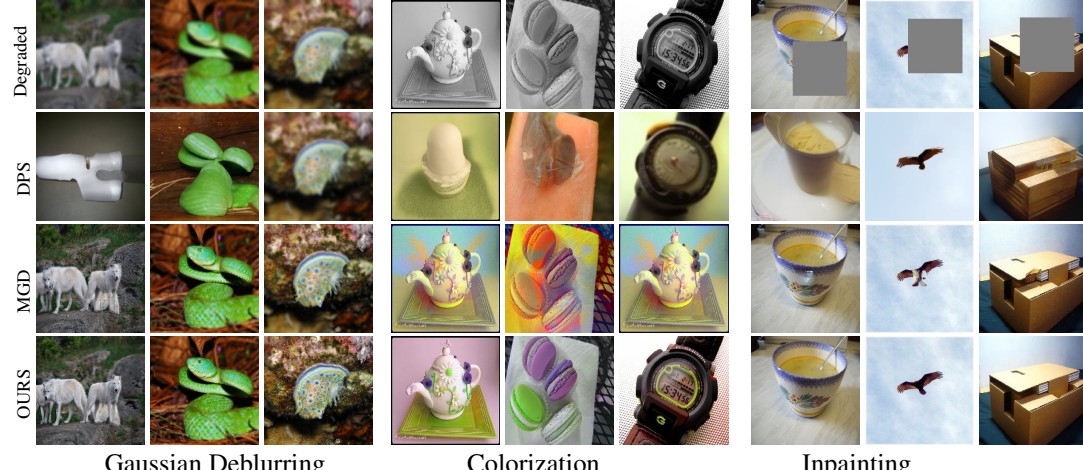

Figure 3: Qualitative comparisons for Linear Tasks on ImageNet for 100 inference steps

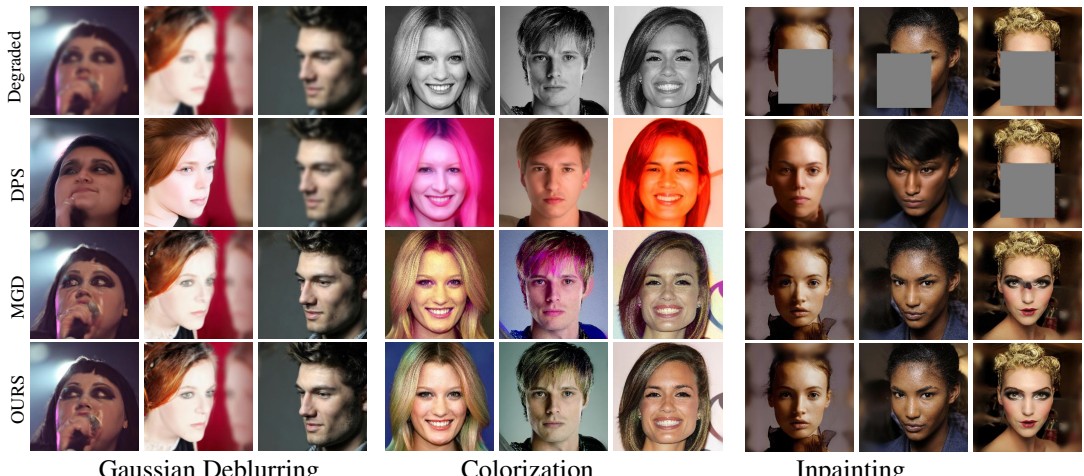

Figure 4: Qualitative comparisons for Linear Tasks on CelebA dataset for 100 inference steps

for Face Sketch guidance, Face Parse maps guidance, and Face ID guidance. Since our method falls into the category of loss-guided diffusion models, we perform all quantitative evaluations using existing methods that follow this kind of sampling. Please note that although we acknowledge the parallel field of research in tackling inverse problems without backpropagation Wang et al. (2023); Kawar et al. (2021), we excluded these methods for comparison as they tackle solely Linear inverse problems. In contrast, loss-guided models are generic and applicable to a wider range of problems.

## 4.1 IMPLEMENTATION DETAILS

We perform all experiments on NVIDIA A6000 GPUs. For ImageNet Deng et al. (2009) based tasks, we utilize the unconditional model released by Guided Diffusion. For Linear Tasks involving faces, we use the model trained on the FFHQ dataset Karras et al. (2017) and perform experiments on the CelebA dataset Liu et al. (2015) similar to DPS. For non-linear tasks, we follow Freedom and utilize the model trained unconditionally on the CelebA dataset. We evaluate using conditions derived from existing networks. For the high-resolution results presented in Figure 2, we utilized the class-conditional model of resolution $512 \times 512$ released by Guided Diffusion. For all experiments, we used 100 sampling steps. For style transfer, we utilized Stable Diffusion Rombach et al. (2021) v1.5. Please note that our sampling method is generic, and any sampler can be used. We fix the number of augmentations in DiffuseAugment for all the experiments to 8. For linear inverse problems we set the value of $t_0$ to 5 in Equation (11) to 30 and for linear inverse problems we set $t_0$ to 5

| Method | Inpaint (Box) | | | | Colorization | | | | SR (× 4) | | | | Gaussian Deblur | | | |
|---|---|---|---|---|---|---|---|---|---|---|---|---|---|---|---|---|
| | PSNR ↑ | SSIM ↑ | LPIPS ↓ | FID ↓ | Cons ↑ | SSIM ↑ | LPIPS ↓ | FID ↓ | PSNR ↑ | SSIM ↑ | LPIPS ↓ | FID ↓ | PSNR ↑ | SSIM ↑ | LPIPS ↓ | FID ↓ |
| Score-SDE Song et al. (2021b) | 9.57 | 0.329 | 0.634 | 94.33 | 0.1627 | 0.3996 | 0.6609 | 118.86 | 20.75 | 0.5844 | 0.3851 | 53.22 | 23.39 | 0.632 | 0.361 | 66.81 |
| ILVR Song et al. (2021b) | - | - | - | - | - | - | - | - | 26.14 | 0.7403 | 0.2776 | 52.82 | - | - | - | - |
| DPS Chung et al. (2023a) | 19.39 | 0.610 | 0.3766 | 58.89 | 0.0069 | 0.5404 | 0.5594 | 55.61 | 17.36 | 0.4969 | 0.4613 | 56.08 | 20.52 | 0.5824 | 0.3756 | 52.64 |
| MGD Chung et al. (2023a) | 27.21 | 0.7460 | 0.2197 | 11.83 | 0.0018 | 0.6865 | 0.4549 | 38.22 | 27.51 | 0.7852 | 0.2464 | 60.21 | 27.23 | **0.7695** | 0.2327 | 51.59 |
| Ours | **28.84** | **0.8491** | **0.1432** | **5.96** | **0.0014** | **0.7775** | **0.3036** | **20.89** | **29.47** | **0.8429** | **0.1757** | **46.95** | **27.30** | 0.7672 | **0.2202** | **42.70** |

Table 2: Quantitative evaluation of image restoration tasks on CelebA 256×256-1k with $\sigma_y = 0.05$, We utilize 100 inference steps for all methods

| Method | Inpaint (Box) | | | | Colorization | | | | SR (× 4) | | | | Gaussian Deblur | | | |
|---|---|---|---|---|---|---|---|---|---|---|---|---|---|---|---|---|
| | PSNR ↑ | SSIM ↑ | LPIPS ↓ | FID ↓ | Cons ↑ | SSIM ↑ | LPIPS ↓ | FID ↓ | PSNR ↑ | SSIM ↑ | LPIPS ↓ | FID ↓ | PSNR ↑ | SSIM ↑ | LPIPS ↓ | FID ↓ |
| Score-SDE Song et al. (2021b) | 9.66 | 0.2087 | 0.7375 | 133.54 | 0.1723 | 0.3105 | 0.8197 | 194.87 | 14.07 | 0.2468 | 0.6766 | 129.91 | 15.39 | 0.3158 | 0.620 | 134.67 |
| ILVR Song et al. (2021b) | - | - | - | - | - | - | - | - | 15.51 | 0.4033 | 0.5253 | 64.13 | - | - | - | - |
| DPS Chung et al. (2023a) | 15.23 | 0.4261 | 0.6087 | 97.90 | 0.021 | 0.3774 | 0.8011 | 106.25 | 14.94 | 0.3258 | 0.6594 | 87.26 | 17.19 | 0.3980 | 0.5817 | 84.74 |
| MGD Chung et al. (2023a) | 21.94 | 0.6920 | 0.2410 | 40.30 | 0.0057 | 0.5809 | 0.5427 | 73.75 | 23.12 | 0.6025 | 0.3936 | 70.83 | 23.13 | 0.6092 | 0.3695 | 61.49 |
| Ours | **23.49** | **0.7271** | **0.2001** | **30.72** | **0.0055** | **0.6804** | **0.3362** | **52.76** | **24.23** | **0.6818** | **0.2884** | **43.00** | **23.31** | **0.6157** | **0.3566** | 58.38 |

Table 3: Quantitative evaluation of image restoration tasks on ImageNet 256×256-1k with $\sigma_y = 0.05$. **Bold**: best, We utilize 100 inference steps for all methods

## 4.2 QUALITATIVE ANALYSIS

We present results on Gaussian Deblurring, super-resolution, and colorization. As we can see, DPS fails since 100 steps of diffusion are used, and the DPS scaling factor is not strong enough to perform proper guidance within 100 steps of diffusion. We set the amount of posterior noise for the measurement as 0.05 in all experiments. MGD works remarkably well for the deblurring and inpainting tasks; however, it fails for colorization since early guidance is required for the flow of natural colors.

For ImageNet tasks, the performance of DPS falls more because the problem is more ill-posed. This can be seen in the eagle diagram, where the method is unable to reconstruct the eagle properly. In contrast, our method performs relatively better, producing much more realistic images. We highlight the performance improvement on colorization since we argue that these results are obtained because of the early flow of gradients. For non linear invere problems, as we can see, Freedom is able to produce realistic-looking results for even the difficult task of Parse Maps to Faces. We argue that this is because backpropagation through the UNet purifies the gradient flow; hence, the generated images look much more naturalistic.

## 4.3 QUANTITATIVE ANALYSIS

We utilize Dreamguider and quantitatively evaluate CelebA and ImageNet datasets. The results for face restoration tasks are shown in Table 2 and 3. We evaluate these tasks utilizing four different metrics. SDEdit Meng et al. (2021) fails for the task of face inpainting and colorization as a single perturbation in the noisy domain throws the image off the manifold. DPS requires more inference steps for proper guidance. ILVR is originally designed for super-resolution. Hence, we quantitatively evaluate ILVR Choi et al. (2021) only for the task of super-resolution. Since DPS and MGD are applicable to all cases, we evaluate with these methods. As we can see, our approach obtains better results than the baselines because of the flow of gradients, which allows for better reconstruction quality. For faces, the difference is much more highlighted in the task of colorization, where we get a significant boost of 18 FID score above the baseline. General linear inverse problems in ImageNet are much more complex than in faces; hence, there is an overall drop in metrics for the natural domain images in ImageNet. In our case, DiffAugment purifies the gradient; hence, we look for much better realistic-looking images. However, MGD does not produce realistic results for sketch-to-image and anime-to-face synthesis.

## 5 ABLATION STUDIES

We perform extensive ablation studies with respect to the effect of DiffuseAugment as well as the effect of each guidance term. For the ablation experiments, rather than utilizing the whole testing dataset of 1000 images, we utilize 100 images and report the average LPIPS value.

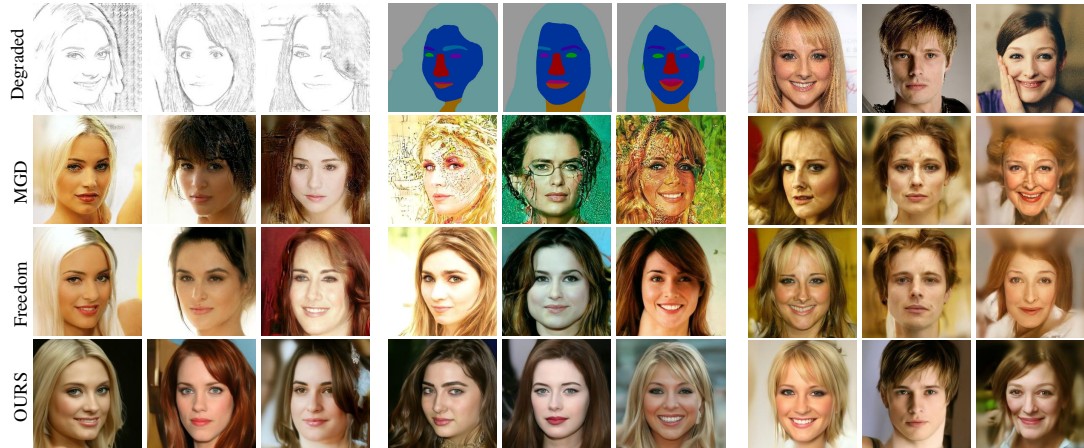

Face-sketch guidance     Face-parse guidance     ID guidance

Figure 5: Qualitative comparisons for Non-linear Tasks on CelebA dataset for 100 inference steps

| Method | Semantic Parsing | | | ID Guidance | | | Face Sketch | | |
|---|---|---|---|---|---|---|---|---|---|
| | Distance↓ | LPIPS↓ | FID↓ | Distance↓ | LPIPS↓ | FID↓ | Distance↓ | LPIPS↓ | FID↓ |
| | | | | *First-order* | | | | | |
| Freedom Yu et al. (2023) | 1864.51 | 0.6030 | 66.89 | 0.3767 | 0.7058 | 81.40 | 39.05 | 0.6583 | 86.51 |
| | | | | *Zeroth-order* | | | | | |
| MGD He et al. (2023) | **2698.27** | 0.6995 | 104.32 | 0.4291 | 0.7178 | 92.61 | 39.34 | 0.6576 | 70.42 |
| Ours | 2722.51 | **0.6199** | **79.42** | **0.3780** | **0.5932** | **82.70** | **39.03** | **0.5509** | **69.51** |

Table 4: Non-linear tasks. Best results out of zeroth-order optimization algorithms are highlighted.

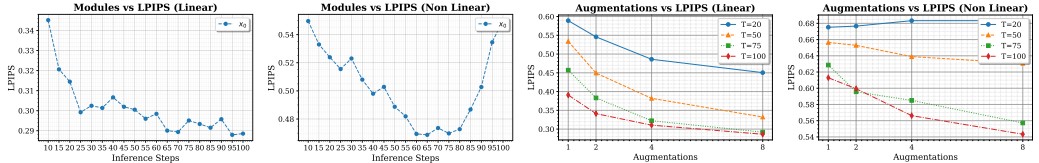

Figure 6: Ablation analysis on linear and non-linear tasks. FaceID guidance & ImageNet superresolution

## 5.1 EFFECT OF DIFFUSEAUGMENT

We notice that for linear tasks, even for low values of $T$ such as $T = 20$, just by increasing the number of augmentations at the output to 8, the perceptual quality drastically improves, matching that of diffusion inference with $T = 50$ with just 2 augmentations. Further, we notice that although the effect of augmentations is very significant for linear tasks, the performance is not that significant or rather drops in some cases for low $T$ such as $T = 20$; this is because with 20 diffusion steps, most intermediate MMSE estimates remain noisy, and hence the guidance network ArcFace Deng et al. (2019) cannot handle such input and hence returns irregular gradients affecting the quality. However, we can see that as $T$ increases and when there are enough gradient steps, DiffuseAugment plays a significant role in boosting the performance.

## 5.2 EFFECT OF DIFFERENT COMPONENTS OF GUIDANCE

We present the ablation analysis of the effect of different terms of guidance in Figure 6. Please note that for this experiment, we set the number of augmentations from DiffuseAugment as 1. We also turn off time travel sampling for this experiment. For this experiment, we perform guidance with respect to $\epsilon_\theta(t)$ until $t_0$ and perform guidance with respect to $\hat{x}_t$ for $t > t_0$. Here $t = 100$ represents pure gaussian noise and $t = 0$ represents the image. As we can see, guidance with $\hat{x}_t$ alone faces a drop in performance initially for a low number of inference steps for non linear cases. We argue that this is because the guidance flow through the MMSE estimate is weak during the earlier steps of diffusion. Although time travel sampling helps to alleviate this issue, careful parameter tuning is

required to obtain satisfactory results. We also notice that guiding utilizing the gradients of the output noise of the network closer to the start of the generation process produces better results.

## 6 LIMITATIONS AND FUTURE WORKS

Although we illustrated the working across various tasks for pixel space diffusion models, the direct approach cannot be used for latent diffusion models for the task of linear inverse problems, and one might have to apply multiple steps of time travel sampling to fix this issue, making a large computational overhead of the overall sampling time. We emphasize that this problem arises due to the reconstruction error in the VAE that encodes the image to the latent space. In the future, we will attempt to improve upon this with better optimization techniques. Moreover, although the proposed empirical estimate based on distance over gradients works for most tasks and shows the existence of an optimal parameter estimate, a thorough mathematical evaluation and the most optimal parameters are still missing. We leave this problem up to future works to estimate the optimal guidance parameter.

## 7 CONCLUSION

In this paper, we proposed an improvement to existing loss-guided techniques for zero-shot conditional generation with an unconditional diffusion model. Specifically, we proposed a sampling technique that removes the need to backpropagate through the diffusion U-Net in order to tackle sampling for general inverse problems. We also present an empirical function for automatic scaling parameters that removes the need for manual scaling parameter tuning, which was previously a huge hurdle in using classifier-free guidance. The newly proposed scaling parameter also removes the need for model-specific tuning of start and end guidance steps. We also introduced a differentiable data augmentation method that significantly improves the sampling fidelity. We illustrated the working of our method across 4 linear and 3 non-linear tasks across faces and real image domains. Our sampling technique produces photorealistic samples with much lower sampling time and higher fidelity than existing methods.

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

## A    Appendix

## B    Algorithm of Dreamguider

We present the over algorithm of dreamguider without time travel sampling and the parameter estimation algorithm in Algorithm 1

## C    Proof for perturbed Markovian kernel equation

In the main paper, we emphasized that any positive distance function can be utilized for performing conditional generation using the perturbed Markovian kernel equation. Here we proceed to derive the perturbed transition step. For the proof we closely follow the work from Dickenson et al Sohl-Dickstein et al. (2015). Given a unconditional transition distribution $p_\theta(x_{t-1}|x_t)$ and a distance function $r(.,y)$, where y is the condition provided Please note that we assume $r(.,y)$ has relatively small variance compared to $p_\theta(x_{t-1}|x_t)$, We know that at equilibrium state, the distribution at any timestep $t$ ina diffusion model can be written as

$$p(x_{t-1}) = \int p(x_t)p_\theta(x_{t-1}|x_t)dx_t. \tag{16}$$

To estimate a perturbed transition kernel $\hat{p}(x_{t-1}|x_t)$,we start the perturbed distribution as

$$p(x_{t-1})r(x_{t-1},y) = \int r(x_t,y)p(x_t)\hat{p}_\theta(x_{t-1}|x_t)dx_t. \tag{17}$$

By simple algebraic manipulations, taking $r(x_{t-1},y)$ to the other side, we get

$$p(x_{t-1}) = \int \frac{r(x_t,y)}{r(x_{t-1},y)}p(x_t)\hat{p}_\theta(x_{t-1}|x_t)dx_t. \tag{18}$$

By comparing Equation (16) and Equation (18) we can see that one solution for the transitional distribution is

$$\hat{p}_\theta(x_{t-1}|x_t) = p_\theta(x_{t-1}|x_t)\frac{r(x_{t-1},y)}{r(x_t,y)}. \tag{19}$$

Also since normalization constants doesn't affect the score function or transition step, Absorbing $x_t$ to the normalization factor of $p_\theta(x_{t-1}|x_t)$, another valid perturbed transition kernel is

702
703
704
705
706
707
708
709
710
711
712
713
714
715
716
717
718
719
720
721
722
723
724
725
726
727
728
729
730
731
732
733
734
735
736
737
738
739
740
741
742
743
744
745
746
747
748
749
750
751
752
753
754
755

---

**Algorithm 1** Dreamguider

---

**Input:** distance function $r(.,.y)$, condition $y$ , Timesteps $T$

1: $x_T \sim \mathcal{N}(x_T; 0, I)$
2: **for** $t = T - 1, \ldots, 1$ **do**
3: $\quad \Sigma = \sqrt{1 - \bar{\alpha}_t}$
4: $\quad \epsilon \sim \mathcal{N}(\epsilon; 0, I)$
5: $\quad \hat{x}_t = \frac{x_t - \sqrt{1-\bar{\alpha}_t}\epsilon_\theta(x_t)}{\sqrt{\bar{\alpha}_t}}$
6: $\quad$ Compute $\frac{dr(\hat{x}_t, y)}{d\hat{x}_t}$, $\frac{dr(\hat{x}_t, y)}{d\epsilon_\theta(x_t)}$
7: $\quad$ update $c = ESTIMATE(t, \epsilon_\theta(x_t), \frac{dr(\hat{x}_t, y)}{d\epsilon_\theta(x_t)})$
8: $\quad$ update $d = ESTIMATE(t, \hat{x}_t, \frac{dr(\hat{x}_t, y)}{d\hat{x}_t})$
9: $\quad c_t = c\sqrt{\alpha_{t-1}}$
10: $\quad d_t = -d.\frac{1-\alpha_t}{\sqrt{\alpha_t}\sqrt{1-\bar{\alpha}_t}}$
11: $\quad$ **if** $t < t_0$ **then**
12: $\quad\quad x_{t-1} = \frac{1}{\sqrt{\alpha_t}}\left(x_t - \frac{1-\alpha_t}{\sqrt{1-\bar{\alpha}_t}}\epsilon_\theta(x_t)\right) + \sigma_t\epsilon - d_t\Sigma\frac{dr(\hat{x}_t, y)}{d\epsilon_\theta(x_t)}$
13: $\quad$ **else**
14: $\quad\quad x_{t-1} = \frac{1}{\sqrt{\alpha_t}}\left(x_t - \frac{1-\alpha_t}{\sqrt{1-\bar{\alpha}_t}}\epsilon_\theta(x_t)\right) + \sigma_t\epsilon - c_t\Sigma\frac{dr(\hat{x}_t, y)}{d\hat{x}_t} -$
15: $\quad$ **end if**
16: **end for**
17: **function** ESTIMATE($t, f_i, g_t$)
18: $\quad$ **if** $t = T$ **then**
19: $\quad\quad \gamma_t = \frac{1e^{-5}}{\sqrt{g_T^2}}$
20: $\quad\quad$ Store $f_T$,
21: $\quad$ **else**
22: $\quad\quad \gamma_t = \frac{\max i > t |f_i - f_T|}{\sqrt{\Sigma_{i=i}^T g_t^2}}$
23: $\quad$ **end if**
24: $\quad$ Store $\sqrt{\Sigma_{i=i}^T g_t^2}$
25: $\quad$ **return** $\gamma_t$
26: **end function return** $x_0$

---

$$\hat{p}_\theta(x_{t-1}|x_t) = p_\theta(x_{t-1}|x_t)\frac{r(x_{t-1}, y)}{Z}. \tag{20}$$

Please note that the term $Z$ does not affect the transition step in the reverse process when the variance of $r(., y)$ is small.

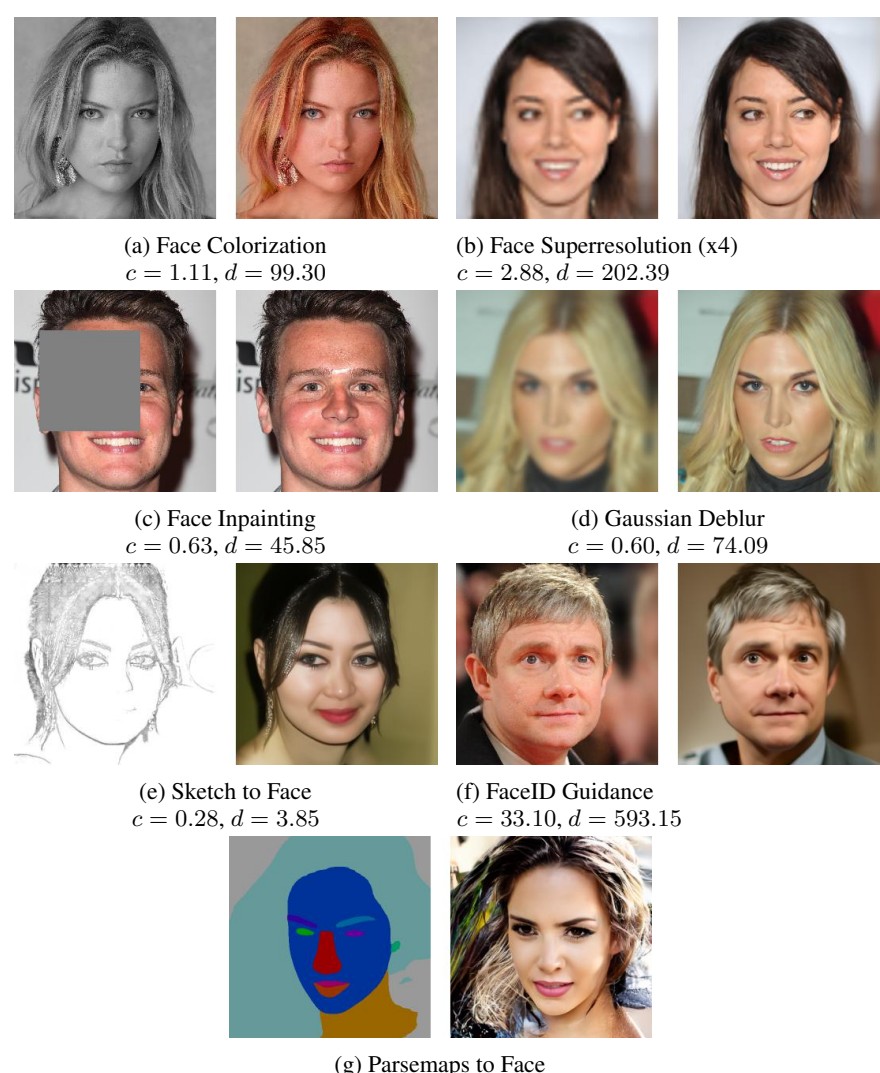

(a) Face Colorization
$c = 1.11, d = 99.30$

(b) Face Superresolution (x4)
$c = 2.88, d = 202.39$

(c) Face Inpainting
$c = 0.63, d = 45.85$

(d) Gaussian Deblur
$c = 0.60, d = 74.09$

(e) Sketch to Face
$c = 0.28, d = 3.85$

(f) FaceID Guidance
$c = 33.10, d = 593.15$

(g) Parsemaps to Face
$c = 0.001, d = 0.13$

Figure 7: Figure illustrating the guidance scales for different tasks.

| Method | Freedom | Dreamguider(1) | Dreamguider(2) | Dreamguider(3) |
|---|---|---|---|---|
| Sketch to Face | 24.95 | 17.55 | 27.04 | 35.09 |
| FaceID to Face | 24.94 | 20.45 | 31.89 | 41.80 |
| FaceParse to Face | 56.25 | 48.35 | 75.43 | 107.02 |

Table 5: Non-linear tasks ablation analysis on time taken, the value is represented in seconds

## D TIME COMPARISON FOR DREAMGUIDER WITH TIMETRAVEL SAMPLING AND FREEDOM(FIRST ORDER) FOR NON LINEAR TASKS

We present the time taken by Freedom, a first order algorithm for one step of time travel sampling Lugmayr et al. (2022); Yu et al. (2023) in Table 5

## E  ESTIMATED PARAMETER VALUE FOR DIFFERENT TASKS

In this section, we present the result and the parameter estimated by our approach for different tasks. For this experiment, we use 100 timesteps of diffusion and present the value at the 100th timestep. Here we define $d$ as the scaling factor of the scaling constant of the the loss derivative relative to $\epsilon_\theta(x_t)$ and c as that of $\hat{x}_t$ as in the main paper . The corresponding results are shown in Figure 7

## F  NON CHERRY PICKED RESULTS FOR DIFFERENT TASKS.

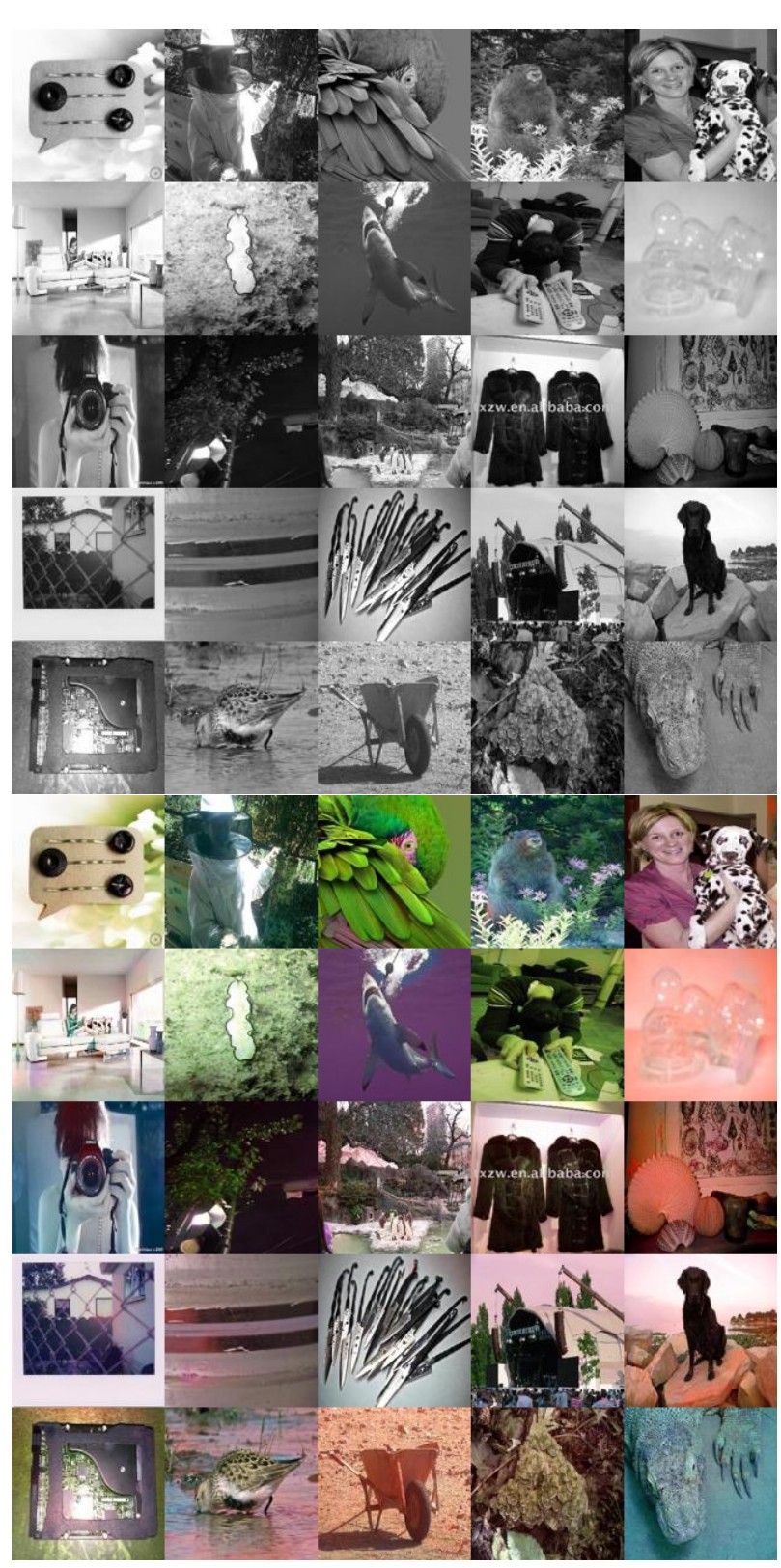

Figure 8: Figure illustrating **Non cherry picked** results for ImageNet colorization

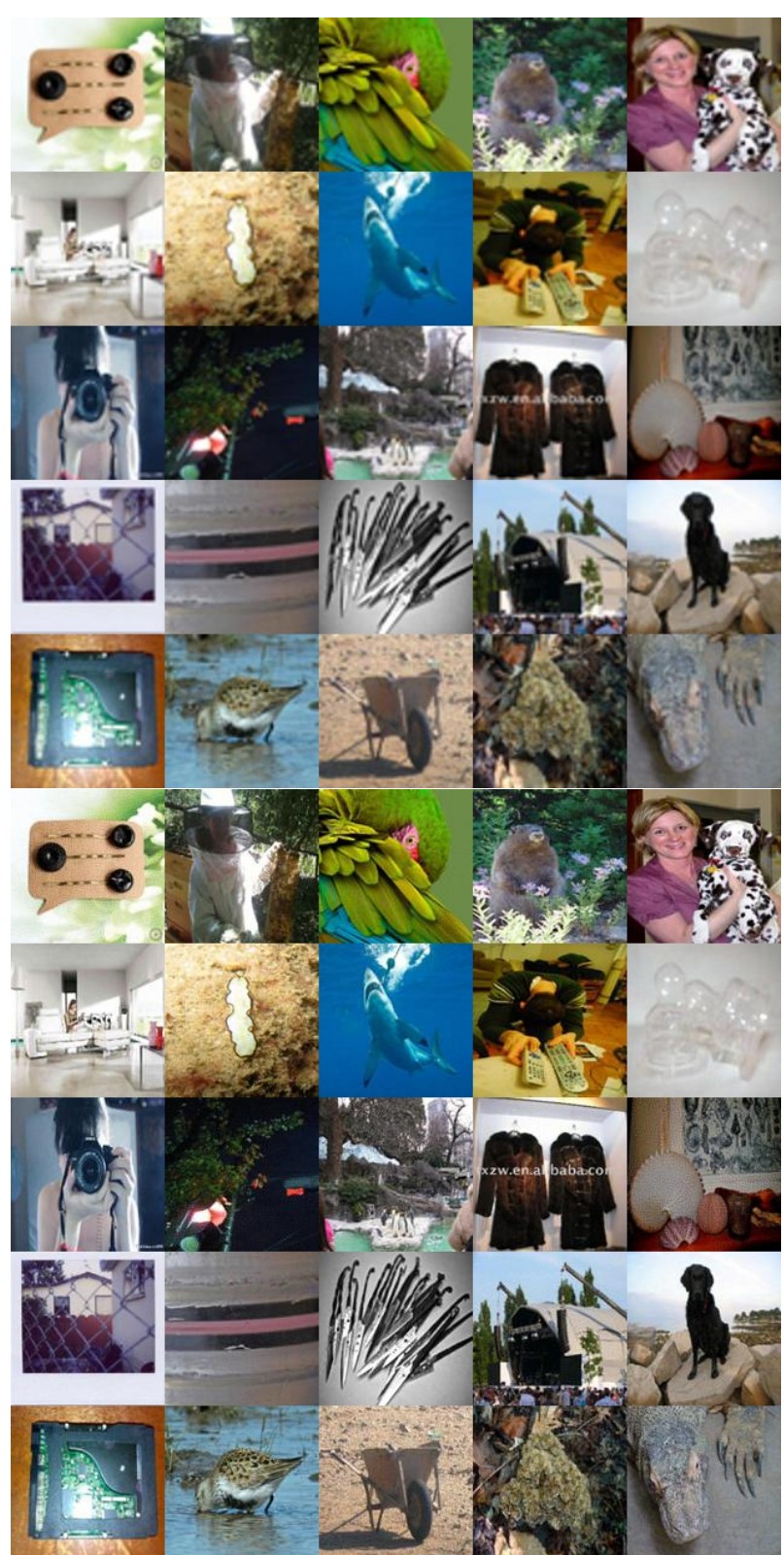

Figure 9: Figure illustrating **Non cherry picked** results for ImageNet superresolution

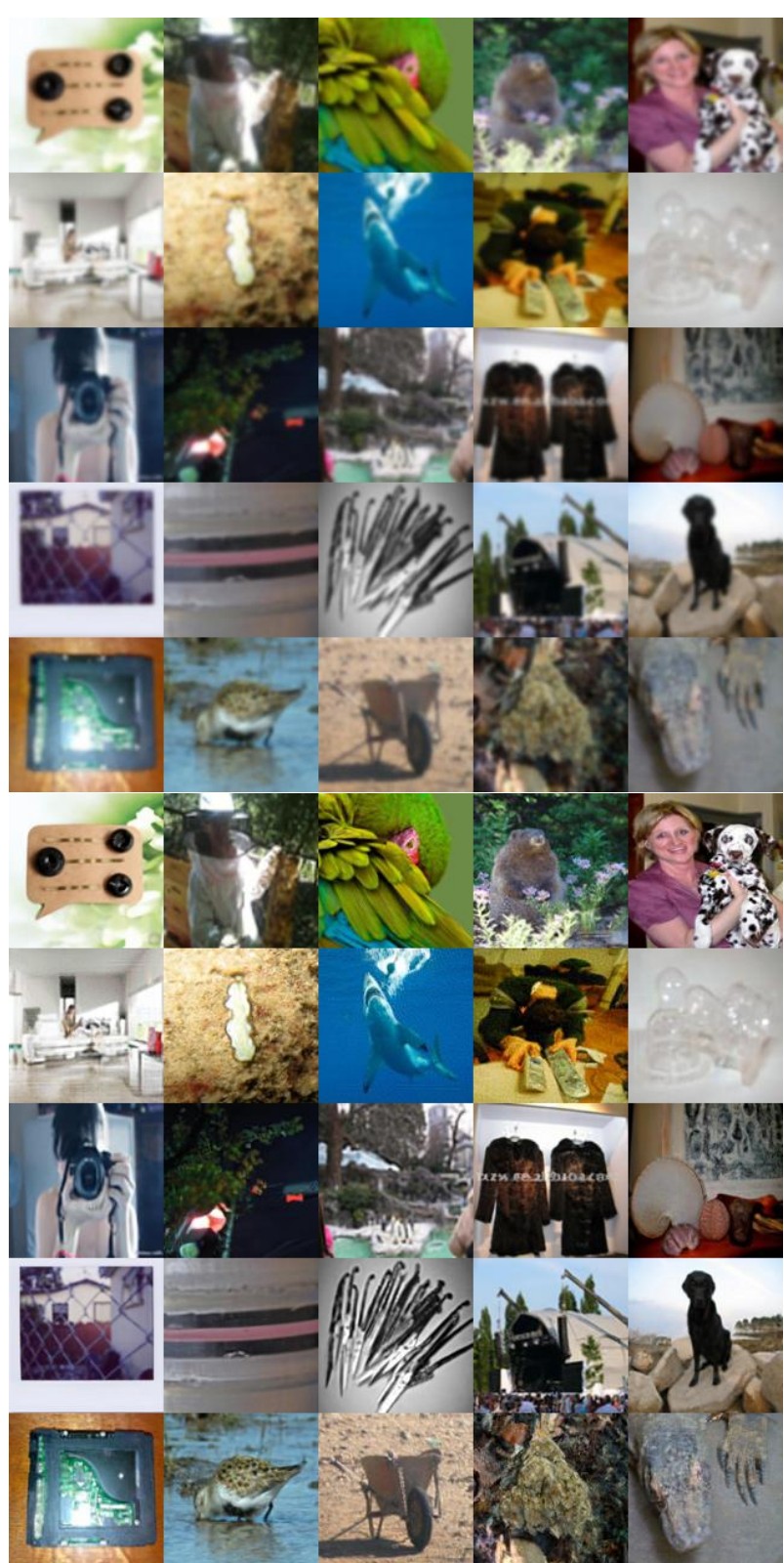

Figure 10: Figure illustrating **Non cherry picked** results for Gaussian deblurring on ImageNet

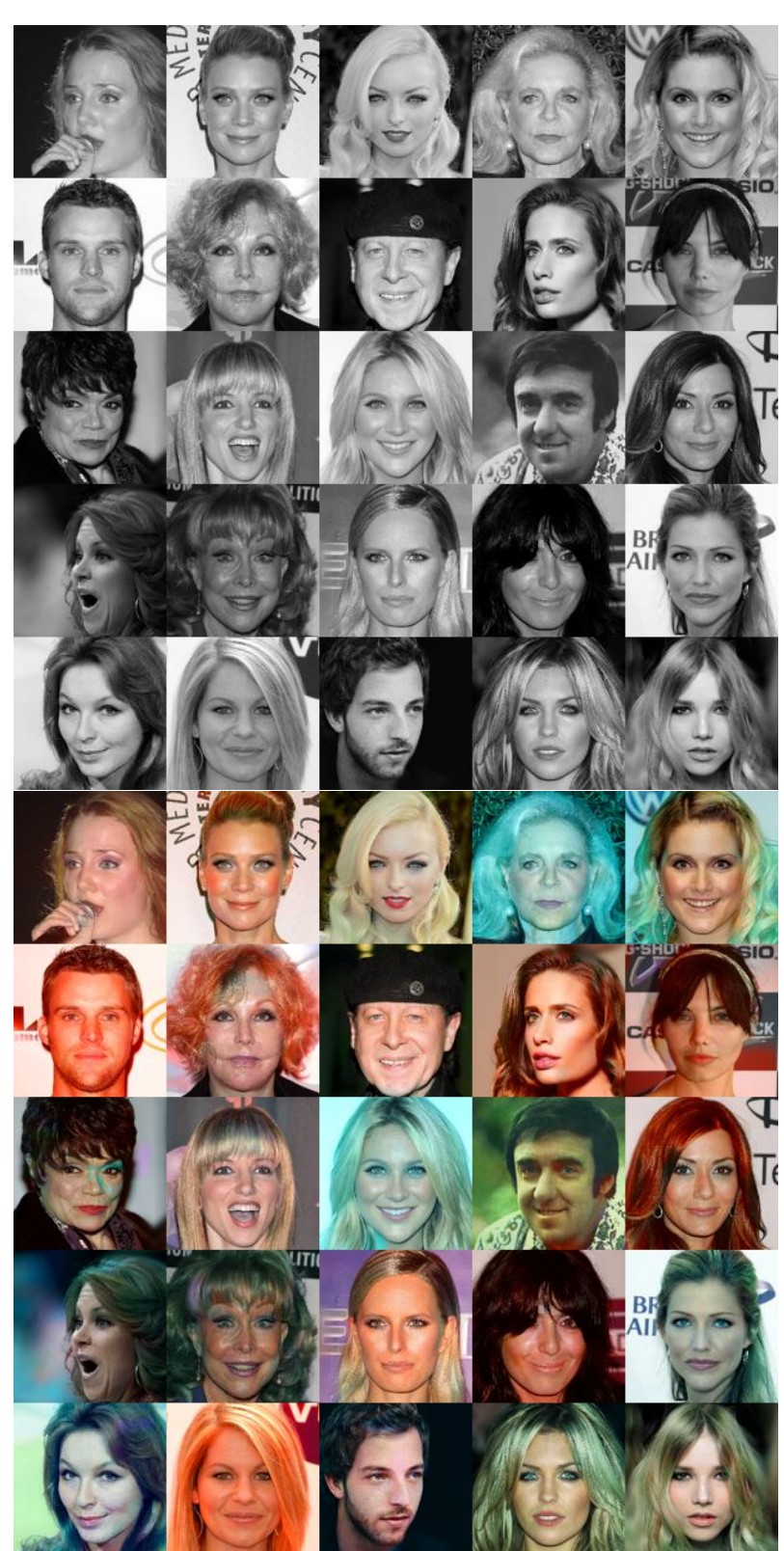

Figure 11: Figure illustrating **Non cherry picked** results for face colorization

1080
1081
1082
1083
1084
1085
1086
1087
1088
1089
1090
1091
1092
1093
1094
1095
1096
1097
1098
1099
1100
1101
1102
1103
1104
1105
1106
1107
1108
1109
1110
1111
1112
1113
1114
1115
1116
1117
1118
1119
1120
1121
1122
1123
1124
1125
1126
1127
1128
1129
1130
1131
1132
1133

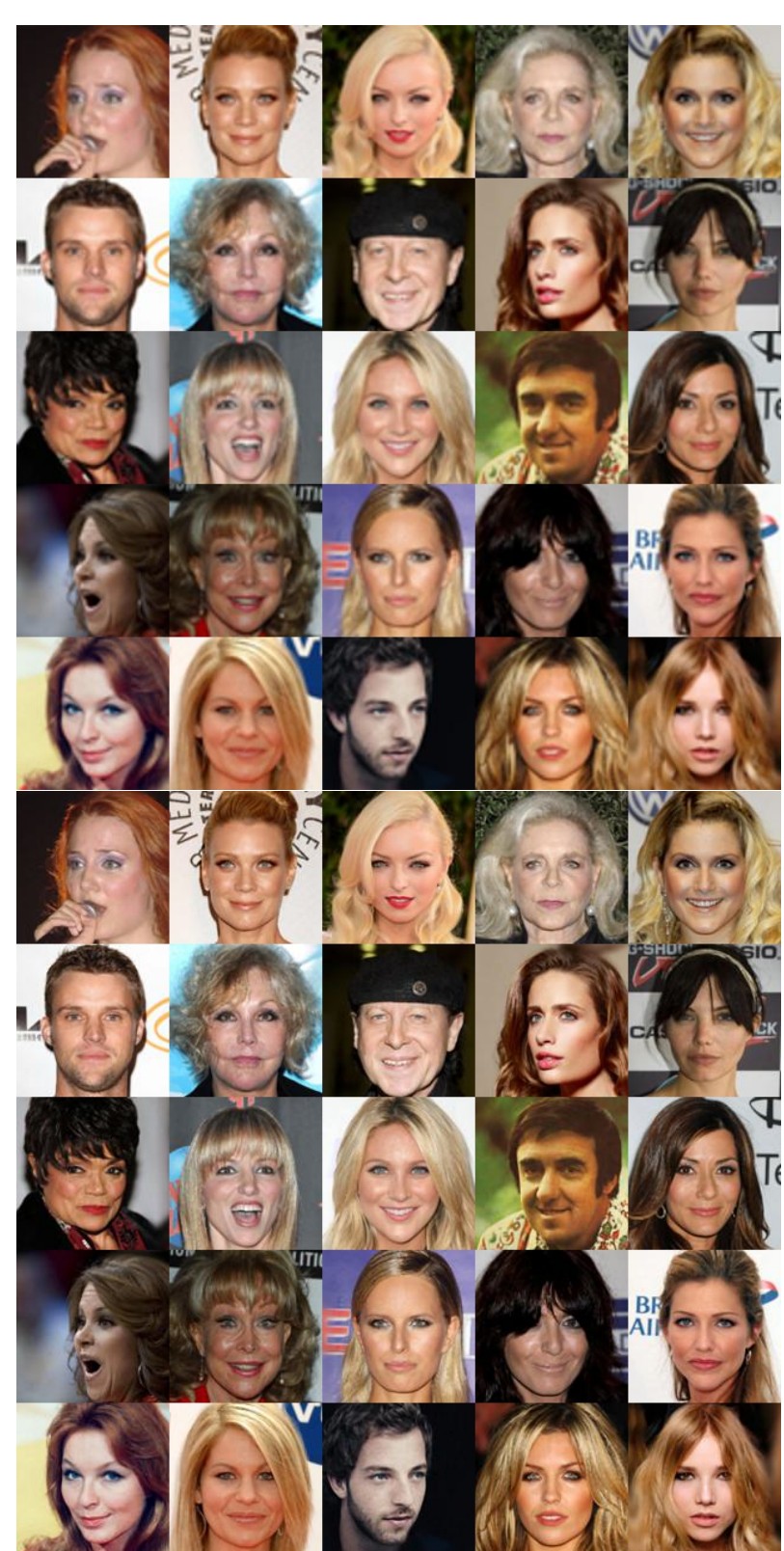

Figure 12: Figure illustrating **Non cherry picked** results for face superresolution

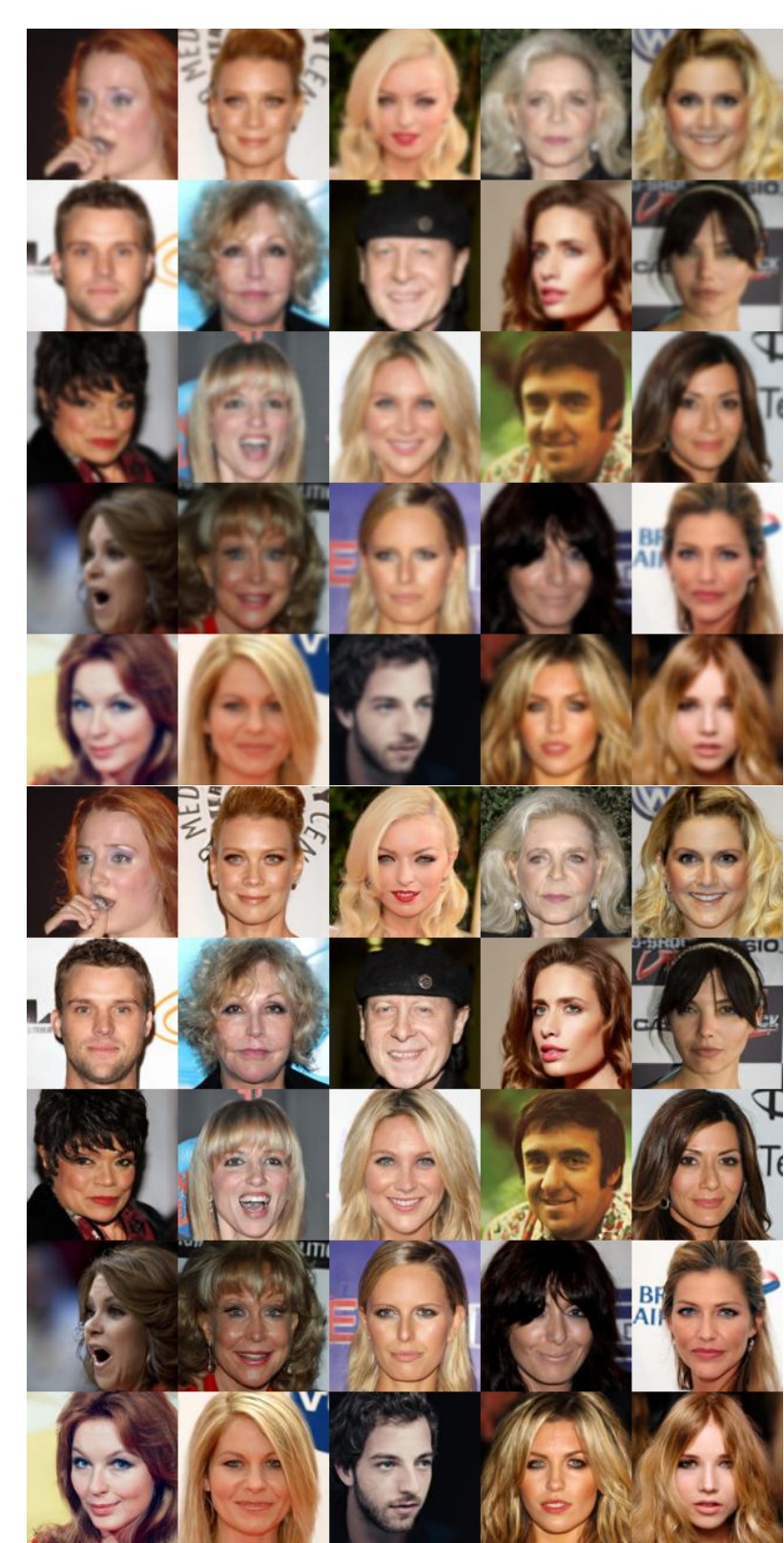

Figure 13: Figure illustrating **Non cherry picked** results for Gaussian Deblurring

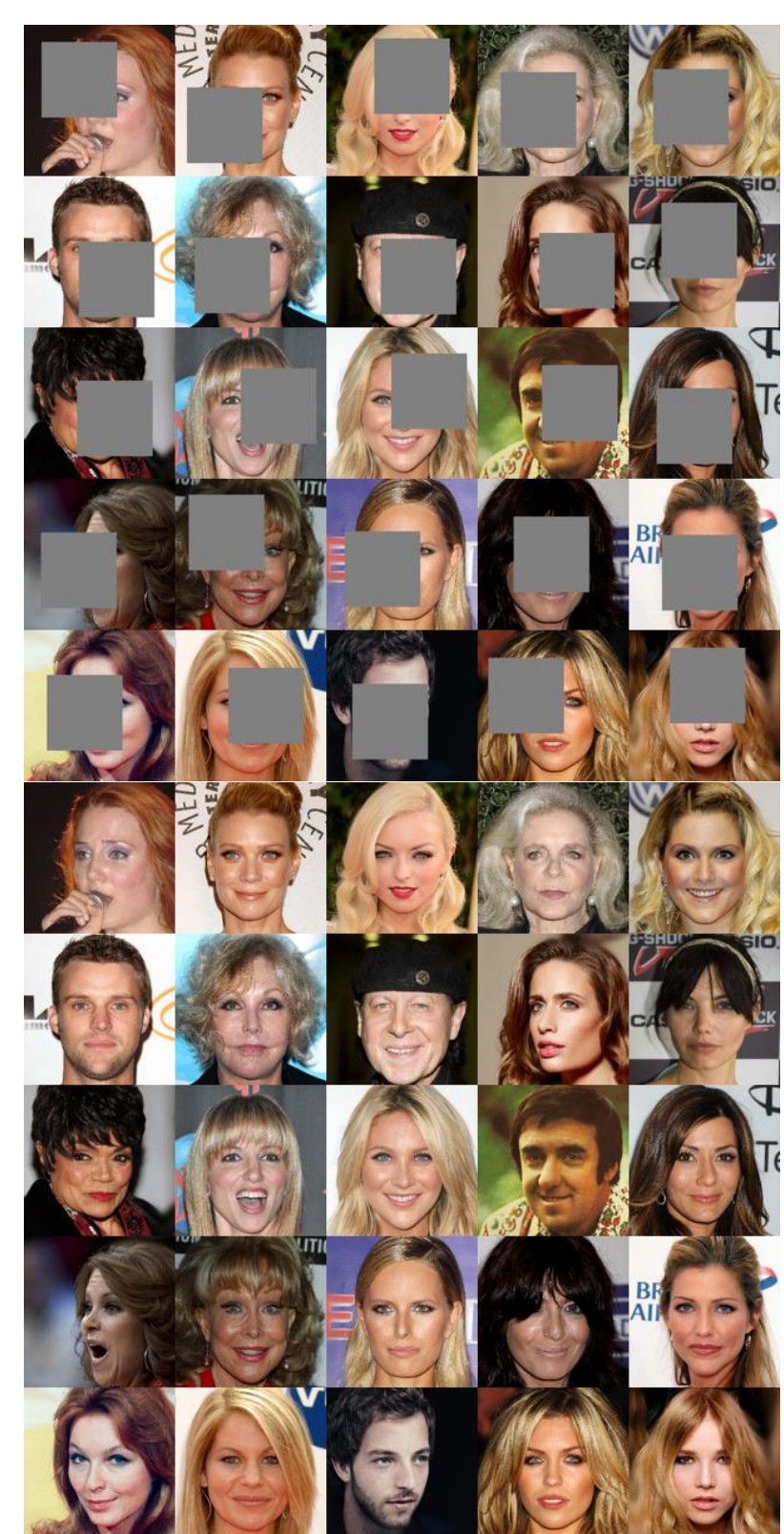

Figure 14: Figure illustrating **Non cherry picked** results for face inpainting

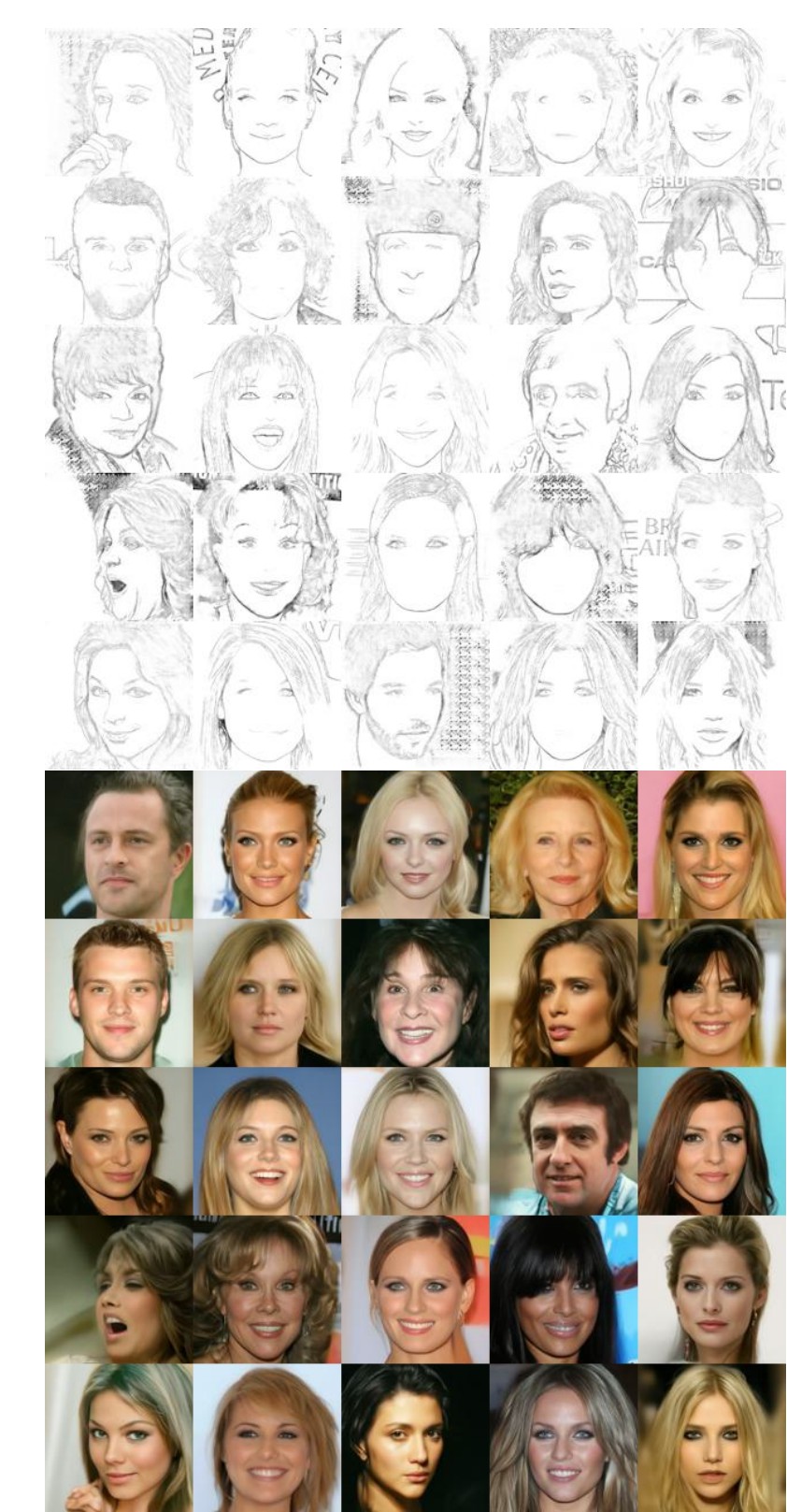

Figure 15: Figure illustrating **Non cherry picked** results for sketch to face synthesis

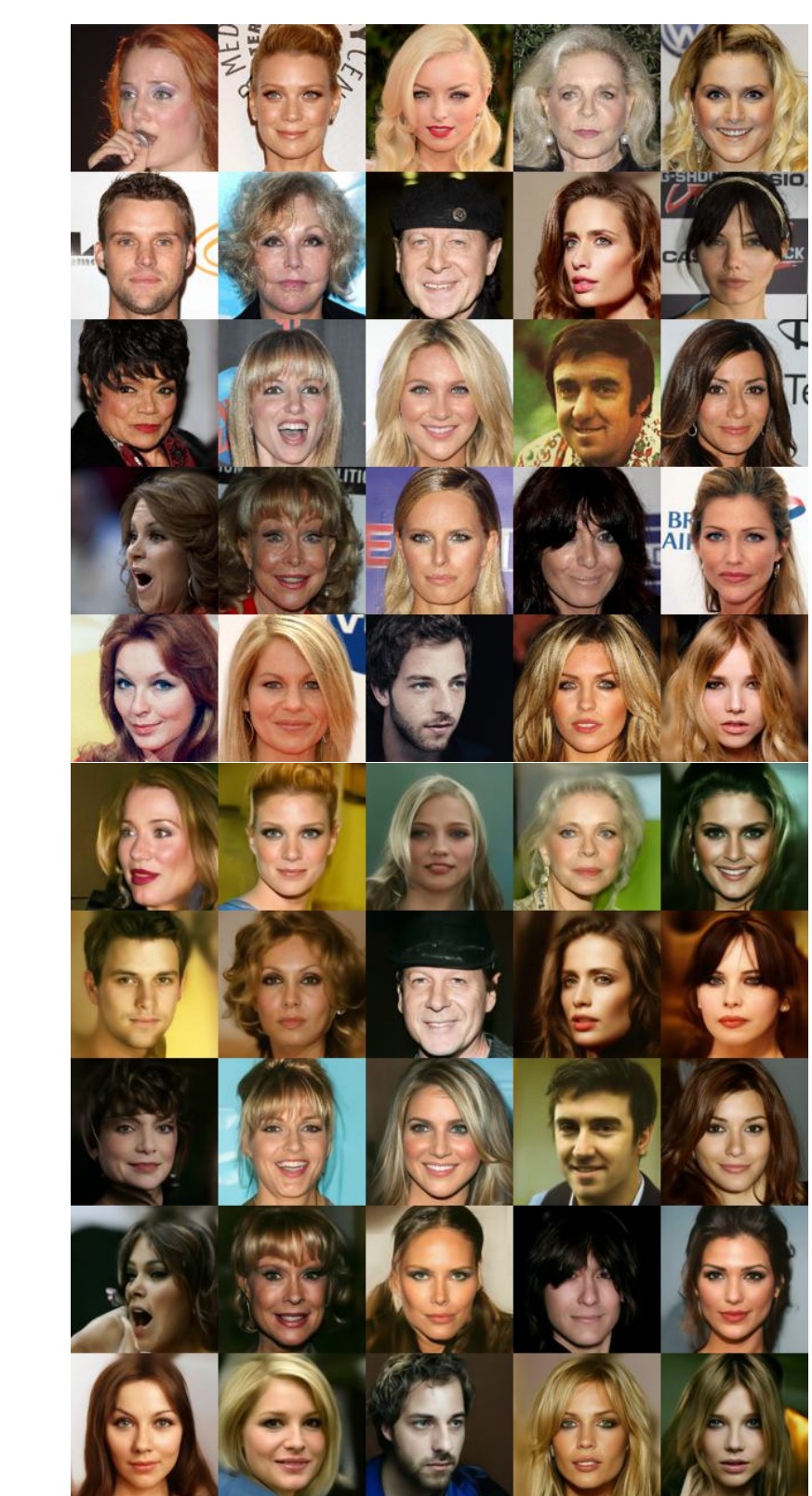

Figure 16: Figure illustrating **Non cherry picked** results for Face ID guidance

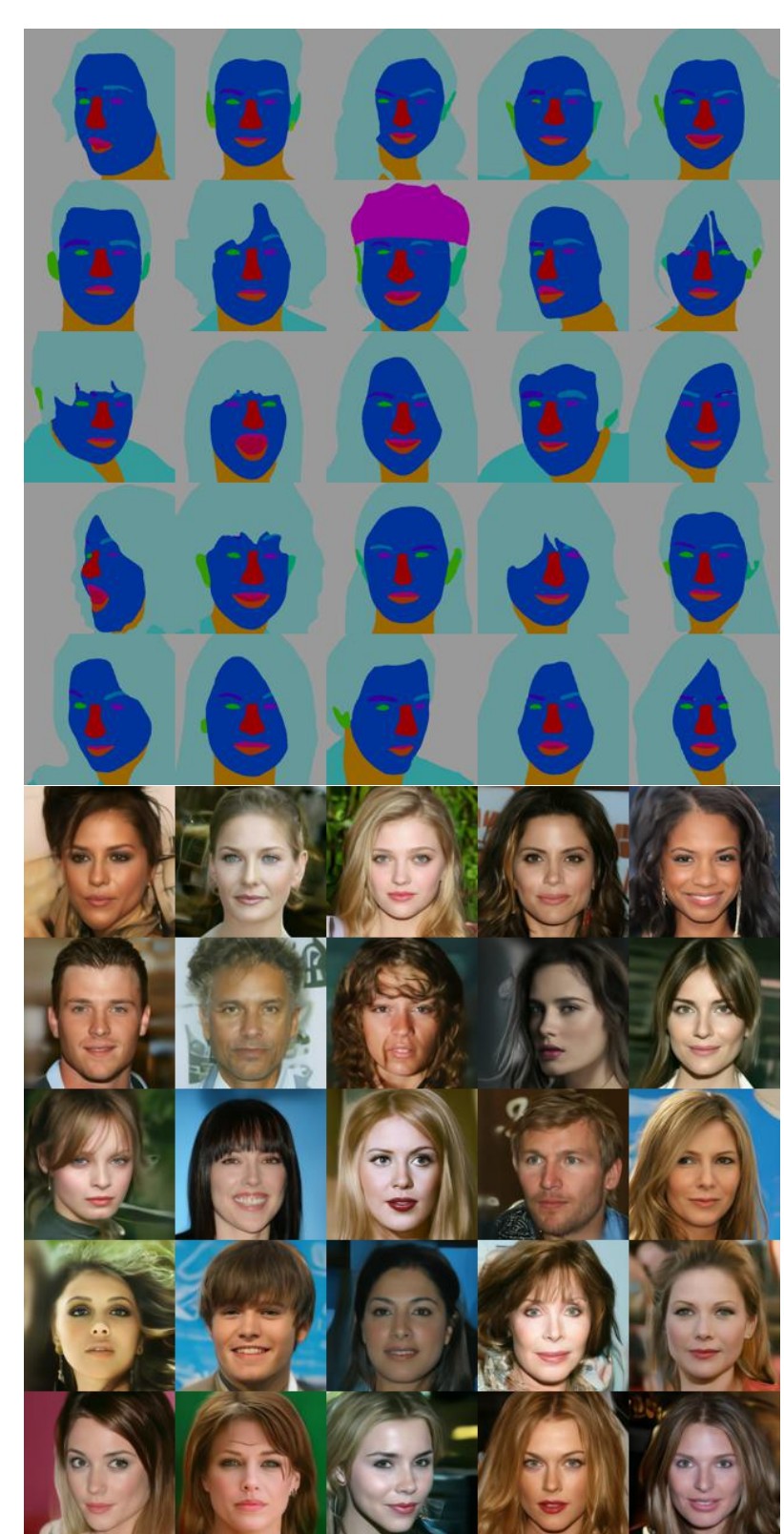

Figure 17: Figure illustrating **Non cherry picked** results for Face Parse Guidance

