# OpenReview forum: "Dreamguider: Improved Training free Diffusion-based Conditional Generation"
_ICLR.cc/2025/Conference — ICLR 2025 Conference Withdrawn Submission_

### Official Review · Reviewer_K76Z · 2024-11-03

**Soundness:** 3
**Presentation:** 3
**Contribution:** 3
**Rating:** 5
**Confidence:** 2

**Summary:**

This paper propose Dreamguider which solve both linear and non-linear inverse problem. While all the previous work requires to contain back propagation path including the diffusion model, this paper find the way to compute MMSE estimates without diffusion network.

**Strengths:**

- The topic looks promising. Solving the inverse problem with less compute is important.
- The performance gap between existing method looks good.

**Weaknesses:**

- The motivation of proposed method is "fast sampling", but speed comparison to existing method is missing for table 2,3,4. I can only see Figure 6 about the speed, but this is self ablation study. I want see the figure that x-axis = speed / y-axis=performance compared to existing method.

- Can you apply this method on "solving inverse problem with unconditional flow-matching model"?

**Questions:**

See weaknesses.

---

> ### Author Response · Authors · 2024-11-12
>
> Thank you for your valuable time and insightful comments! We have tried to address your concerns in the updated manuscript and our rebuttal text:
>
>
> 1. We present the speed of our algorithm compared to a first-order algorithm in Table 5 of the paper. DPS, a first-order algorithm, requires 1000 diffusion steps. The complexity of DPS is similar to that of Freedom, which also uses first-order optimization but with only 50 steps. Therefore, we compare DreamGuider’s speed with Freedom in our evaluations. Even with techniques like time-travel sampling, DreamGuider requires relatively lower time for sampling
>
> 2. Yes, the DreamGuider sampling algorithm can be integrated with unconditional flow-matching as well. We will include the results during the discussion period.

---

### Official Review · Reviewer_dq4y · 2024-11-06

**Soundness:** 3
**Presentation:** 2
**Contribution:** 2
**Rating:** 3
**Confidence:** 4

**Summary:**

The paper proposes a training-free algorithm for conditional inference in diffusion models. The authors improve upon previous works by coming up with a guidance schedule that balances better the conditioning signal during the inference process. They also propose a differentiable augmentation strategy that improves the guidance by averaging it over different augmentations of the input and the condition. They validate their method experimentally on a set of linear and non-linear inverse problems, across different diffusion models.

**Strengths:**

- The proposed algorithm seems to overall improve upon the results attained by previous methods. When comparing with the similar work done in MGD, the results show that there is a clear advantage for the proposed algorithm, both quantitatively and qualitatively.
- The augmentation guidance scheme proposed, could be of wider applicability and improve other training-free conditional inference methods in the future. Since the augmentation is not dependent on the method used it has the potential to be adopted by other works in the future.

**Weaknesses:**

- The novelty is somewhat limited. The proposed algorithm seems to be an extension of the previous work of MGD (also a zeroth-order guidance scheme) and the contributions are not clear enough. Out of the four contributions mentioned in the introduction the ones that are clearly presented are regarding the tuning of the constraint guidance and the differentiable augmentations.
- In Table 2 and Table 3 the citation for MGD is incorrect (points to DPS).

**Questions:**

- What is the main difference between the proposed method and MGD? Both algorithms compute the gradient $\nabla_{\hat{x}_t} r(\hat{x}_t, y)$ and add it to $x\_{t-1}$. The proposed algorithm also adds $\nabla\_{\epsilon\_{\theta}(x\_t)} r(\hat{x}_t, y)$ but $\hat{x}_t$ and $\epsilon\_{\theta}(x\_t)$ are related by Eq. 10. It seems like that the gradient $\nabla\_{\epsilon\_{\theta}(x\_t)} r(\hat{x}_t, y)$ is not providing any extra guidance/information for the inference process but is rather used as a proxy to correctly weigh the change of $x\_{t-1}$ to match the constraint at timestep $t-1$. Given the above, could someone reformulate your method as MGD with different weighting?

---

> ### Author Response · Authors · 2024-11-12
>
> Thank you for your valuable time and insightful comments! We have tried to address your concerns in the updated manuscript and our rebuttal text:
>
>
> 1. **Motivation for Zeroth-Order Scheduled Guidance and how it is different from MGD**:
>    MGD only performs guidance through $\hat{x}\_{t}$, which limits gradient flow for tasks requiring semantic understanding. It has not been evaluated on non-linear, semantic tasks like segmentation map-image generation or sketch-to-image generation, focusing instead on linear inverse problems and style transfer tasks. We believe that semantic features are formed early in the generation process, where guidance through $\hat{x}\_{t}$ is weaker. Conversely, gradient flow is stronger through $\epsilon_{\theta}(x_t)$ at these steps. This observation motivated our proposed zeroth-order scheduled guidance.
>
> 2. **Main Contribution – Automatic Step-Size Scheduling**:
>    Our primary contribution is the design of an automatic, time-dependent step-size schedule. In previous works on diffusion guidance, this parameter has been manually crafted for each case. We propose an empirical estimate that generalizes well across tasks. Please refer to Figure 7 in our paper for a visual representation. We believe this design will allow researchers to apply this technique across diffusion-guided tasks without focusing on hyperparameter tuning, enabling them to concentrate on algorithmic development.
>
> 3. **Stable Gradient Flow with Differentiable Augmentation**:
>    To ensure a stable gradient flow, we propose a differentiable augmentation scheme, which improves performance across a range of tasks.
>
> ---
>
> **Regarding the Reviewer’s Query:**
>
> 1. While it is true that there is only a scaling difference between the gradients with respect to $\epsilon\_{\theta}(x\_t)$ and $\hat{x}\_t$, please note that we perform guidance at different time schedules, not simultaneously. We apply guidance through $\epsilon\_{\theta}(x\_t)$ in the initial diffusion steps, where semantic features form, as guidance through $\hat{x}\_t$ is weaker here. This motivates our scheduled gradient guidance approach, which varies with timesteps.
>
> ---
>
> I hope this helps clarify our approach and contributions. Thank you for your consideration.

---

### Official Review · Reviewer_ab9K · 2024-11-06

**Soundness:** 2
**Presentation:** 2
**Contribution:** 3
**Rating:** 5
**Confidence:** 4

**Summary:**

In this paper, the authors propose a new method for training-free diffusion posterior guidance. In particular, instead of only modifying the Tweedie’s (MMSE) estimation in DDIM sampling, they propose to also guide the noise/score prediction to obtain a better guided sample. They also provide a way to perform data augmentation and a closed form solution to the step size of these guidance, which is a critical contribution to the field as so far there is no principle way to determine these hyperparameters. Their experiments show superior performance in comparison to baselines across many different tasks.

**Strengths:**

Almost all prior works in training-free diffusion guidance are very sensitive to hyperparameters, especially the step size, and currently there is no principled way to determine these hyperparameters. In fact, I think this is one of the biggest problems with these methods that prevent them from being applied widely. This paper provides a concrete way to decide these hyperparameters, which is extremely valuable for this field. The authors also propose a very creative way to perform the guidance by modifying the noise/score estimation directly (as opposed to the Tweedie’s clean data estimation only), which I find to be very interesting as well. The authors also provide a very effective way to perform data augmentation in this diffusion guidance task, which is also very useful.

Overall I really like the methods proposed in this paper, and I really want to accept it. However, there are certain aspects in this paper that I think need to be improved before it gets accepted.

**Weaknesses:**

1. The writing should really be improved: there are a lot of undefined notations and unjustified algorithm components.
2. The main method (Section 3.1 & 3.2) is not very well explained. I think guiding the $\epsilon$ is a very interesting idea and would love to see more theoretical justifications about why this is effective. Section 3.2 is not very informative at all, I would love to see the deduction process of Eq 14.
3. Eq 1 is a very strong assumption, maybe it is better to assume proportional to rather than equals.
4. Also distance (r) doesn’t necessarily implies relationships to $p(y \mid x_t)$, so it may not be connected to Eq 6 and hence you don’t necessarily get Eq 9. What exactly is the assumption made here?
5. $\Sigma$ is not clearly defined when it’s first mentioned in Eq 8, since the authors use Tweedie’s/DDIM (MMSE) estimation for $\hat{x}_t$, is it just $\sqrt{1-\alpha_t}I$?
6. I think Eq 11 (Line 256-261) is quite confusing. Do you change $\hat{x_t}$ and $\epsilon_\theta(x_t)$ and then plug them into the $x_{t-1}$ formula? Or the $x_{t-1}$ formula is a summary of the changes you would do in the $\hat{x_t}$ formula and $\epsilon_\theta(x_t)$ formula? Seems like the latter one is consistent with the algorithm provided in the appendix, but it is very confusing to read here.
7. I think Eq 12 should be $c_t = \frac{c}{\sqrt{\alpha_t}}$ and in Eq 13 $d_t$ shouldn’t have the negative sign?
8. The authors claim that Eq 14 also works very well for DPS, do they have any experimental results to show that?
9. Writing suggestion: I don’t think the formulation in Section 2.3 is very different from DPS if you use the Tweedie’s (MMSE) estimation like in Section 3.1, so maybe just introducing the problem following DPS or MPGD is ok.
10. MPGD can also perform nonlinear tasks (error in Table 1)

Minor:
1. Line 256 -261, it should be partial derivatives $\partial$ instead of $d$.
2. Eq 9 should be $-\Sigma\frac{\partial r(x_{t-1},y) }{\partial x_{t-1}}$ instead of $+$ because usually the smaller the distance is the larger the conditional probability would be. This is probably connected to Weakness 7 the problem with Eq 13 and Weakness 4 the assumptions about the distance function and the conditional probability $p(y \mid x_t)$.
3. The method He et. al (2023) proposed is called MPGD not MGD.
4. Citation error in Line 382 & 389.

I am willing to accept this paper if the authors can improve the writing and provide more justifications as suggested above.

**Questions:**

It would be great if the authors can address the weakness mentioned above.
In addition, I am wondering if the authors have used techniques such as time traveling/repaint to stabilize the sampling process?

---

### Official Review · Reviewer_f7oS · 2024-11-07

**Soundness:** 2
**Presentation:** 1
**Contribution:** 1
**Rating:** 3
**Confidence:** 5

**Summary:**

The authors propose a generic lightweight guidance solution, named Dreamguider, which enables inference-time guidance without the need for backpropagation through the entire diffusion network. Dreamguider can address both linear and non-linear guidance problems. The authors also introduce an empirical guidance scale strategy to removing the need for handcrafted parameter tuning. This paper showcases experiments across multiple tasks, datasets, and models to demonstrate the effectiveness of the proposed methods. The key contributions include: 1) Dreamguider, a zeroth-order loss-guided diffusion guidance applicable to both linear and non-linear inverse problems, 2) a time-varying guidance scale, and 3) a differentiable augmentation strategy.

**Strengths:**

1. The paper tackles a relevant and challenging problem in the realm of conditional (controllable) generation for unconditional diffusion models.
2. It proposed two useful tricks to improve the generation quality: 1) an empirical guidance scale and 2) differentiable augmentation for classifier guidance.

**Weaknesses:**

1. Overall, the contributions of this paper are limited. The proposed Dreamguider is built upon MGD [1], an earlier proposed zeroth-order loss-guided guidance method that also eliminates the need for backpropagation through the diffusion network. The authors claim that Dreamguider can address non-linear inverse problems, unlike MGD. However, they should provide a detailed description and evidence to support this claim. In the current manuscript, I cannot find the intuition or evidence needed to substantiate this.

2. The paper should clarify the intuition behind the methodological design discussed in Section 3.1, which currently lacks depth and clarity. Given that Dreamguider is closely related to MGD, a comprehensive comparison of each designed component should be included to enhance understanding.

3. MGD is capable of handling latent diffusion models, so it is ridiculous that an improved method like Dreamguider loses this important capability. The authors should provide a detailed explanation for this shortcoming, and consider addressing it rather than deferring the issue to future work.

4. The showcased results are not appealing, as the performance differences between Dreamguider and MGD on some tasks are marginal. If possible, incorporating a user study could significantly enhance the experimental validation.

Typos:

In Table 2, some second-rank values are incorrectly bolded.

[1] Yutong He, et al. "Manifold preserving guided diffusion" ICLR 2024.

**Questions:**

Please refer to the weakness for my questions.

---

> ### Author Response · Authors · 2024-11-12
>
> Thank you for your valuable time and insightful comments! We have tried to address your concerns in the updated manuscript and our rebuttal text:
>
> 1. **Motivation for Zeroth-Order Scheduled Guidance and Differences from MGD**:
>    MGD only performs guidance through $\hat{x}\_t$, which limits gradient flow for tasks requiring semantic understanding. It has not been evaluated on non-linear, semantic tasks like segmentation map-image generation or sketch-to-image generation, focusing instead on linear inverse problems and style transfer tasks.
>
>    We believe that semantic features are formed early in the generation process, where guidance through $\hat{x}\_t$ is weaker. Conversely, gradient flow is stronger through $\epsilon\_{\theta}(x\_t)$ at these steps. This observation motivated our proposed zeroth-order scheduled guidance. We will include these details in the revised version, including the guidance gradients at different steps. To illustrate the effectiveness of DreamGuider for non-linear inverse tasks, we present results on sketch-to-image synthesis and semantics-to-image synthesis in Figure 5 of the main paper.
>
> 2. **Component-Wise Improvement in Ablation Studies**:
>    We illustrate the improvement from each component in the ablation studies section. Please note that the case with **Inference steps = 100** in Figures 6a and 6b shows the results of MGD. The effect of augmentations is also presented in Figures 6c and 6d.
>
> 3. **Scope of Tasks Addressed by MGD on Latent Diffusion Models**:
>    We would like to draw attention to the tasks addressed by MGD, which focuses on style-based guidance for latent diffusion models. MGD does not function effectively with tasks requiring pixel-level semantics, such as sketch-to-image or semantics-to-image synthesis, nor does it address linear inverse problems on latent diffusion models.
>
>    We thoroughly studied this limitation, which arises due to gradient vanishing through the VAE. Such tasks can only be addressed by pixel-space models. Moreover, the contributions of DreamGuider are focused on (1) an automatic step-size scheduler, (2) differentiable augmentation, and (3) support for non-linear inverse tasks in pixel space.
>
> 4. **User Study Addition**:
>    We appreciate the reviewer’s suggestion and will update the results to include a user study.

---

### Note · Authors · 2024-11-15

**Comment:**

Dear Reviewers,

We thank you for all the time and effort for reviewing our work and providing suggestions for improvement. We will incorporate these and improve our work.

**Withdrawal Confirmation:**

I have read and agree with the venue's withdrawal policy on behalf of myself and my co-authors.